# Research on the Impact of the Digital Economy on Carbon Pollution Based on the National Big Data Comprehensive Pilot Zone in China

**Mingguang Liu [1] and Gaoyang Li [2,*]**

[1] School of Politics and Public Administration, South China Normal University, Guangzhou 510006, China; 20071077@m.scnu.edu.cn
[2] College of Water Conservancy and Civil Engineering, South China Agricultural University, Guangzhou 510642, China
* Correspondence: ligaoyang@scau.edu.cn

**Abstract:** The digital economy (DE) is increasingly recognized as a principal driver of high-quality economic development. With the dual carbon goals of carbon peak and carbon neutrality, it is essential to urgently discuss whether the DE can also provide an aid to restrain carbon pollution. For this reason, the purpose of this paper is to examine the influence of the DE on urban carbon pollution and its underlying mechanism, utilizing balance panel data for Chinese cities from 2012 to 2021. Additionally, this study adopts the quasi-natural experiment in the National Big Data Comprehensive Pilot Zone to empirically analyze this relationship using the Difference-in-Differences (DID) and mediating effect models. The findings indicate that the DE can significantly decrease carbon pollution, exhibiting a clear carbon pollution reduction effect. These conclusions remain valid even after implementing various robustness tests. This examination of the action mechanism reveals that it is effective for the DE to mitigate urban carbon pollution by enhancing energy efficiency and attracting foreign investment. Further analysis of heterogeneity reveals that it is more effective for the DE to reduce carbon pollution in the non-resource-based cities, large cities, eastern regions, and cities with high urbanization levels as compared to the resource-based cities, small or mega cities, central and western regions, and cities with a low level of urbanization. These findings not only offer robust objective evidence for the positive influence of the DE on carbon pollution reduction, but also present valuable policy implications for minimizing carbon pollution and enhancing environmental quality.

**Keywords:** DE; carbon pollution; action mechanism; heterogeneity

## 1. Introduction

After more than 40 years of reform and opening up, the Chinese economy has achieved remarkable success, with its GDP (gross domestic product) surpassing CNY 100 trillion in 2020 and reaching CNY 114 trillion in 2021 [1]. However, along with this rapid development, China's carbon emissions have increased almost 14 times from 1959 to 2020, making it the largest carbon generator. During the United Nations General Assembly on 22 September 2020, President Xi Jinping declared China would strengthen its national independent contributions by implementing stronger policies and strategies and endeavoring to reach the zenith of $CO_2$ emissions before 2030 and carbon neutrality by 2060 [2,3]. Concurrently, China's DE has experienced incredible growth. In accordance with the White Paper on Global DE (2023), which is released by the China Academy of Information and Communications Technology, DE in China has undergone expeditious development from 2016 to 2022. During this period, the size of the DE has grown by USD 4.1 trillion, with a mean annual growth rate of 14.2% [4]. The incessant development of modern information technology, including big data, blockchain, cloud computing, and artificial intelligence, has played a momentous role in driving China's economic development [5,6]. The DE has actualized an

increasingly crucial strength in stimulating consumption, boosting investments, creating job opportunities, and upgrading industrial structures.

With the increasing influence of the DE on various aspects of society, including economic growth, industrial development, and social life, there has been a growing interest in researching how the DE can empower the environment and contribute to sustainable development. Initially, scholars primarily investigated the influence of the DE on various pollutants, such as industrial sulfide gas, industrial sewer water, and industrial smut [7,8]. Subsequently, attention has shifted towards detecting the connection between the DE and factors such as carbon efficiency, carbon dioxide emissions, air quality, and PM2.5. Of particular interest is the carbon reduction effects of the DE, which has emerged as a prominent focus of research. For instance, Dong F, Hu M, Gao Y et al. (2022) conducted a study using national panel data from 2008 to 2018 and found that the DE significantly mitigated carbon pollution vehemence but concurrently played a part in an increase in per capita carbon emissions. The study further identified that economic growth, financial development, and industrial structure upgrading played a critical intermediary role between the DE and carbon pollution [9]. Wang J., Dong K., Dong X. et al. (2022) implemented a study employing panel data from 30 provinces in China from 2006 to 2017. Their findings implied that the DE had a negative impact on carbon pollution. This negative impact principally occurred through several channels, including the expansion of the tertiary industry, the reduction in the proportion of coal consumption, and the booster of green technology innovation. These factors indirectly contribute to a reduction in carbon dioxide emissions [10]. Cheng Y., Zhang Y., Wang J. et al. (2023) analyzed data from 278 prefecture-level cities in China from 2011 to 2019. Their research indicated that industrial structure and technological innovation are important mechanisms of DE affecting carbon pollution [11]. However, upon reviewing the existing research literature, it becomes evident that there are three main limitations. Firstly, most studies utilize an index system to measure the DE, but there is currently no generally recognized index system available. Consequently, there is a substantial discrepancy in the measurement results of the DE, making it challenging to accurately evaluate the carbon pollution reduction effect of the DE. Secondly, while numerous studies have explored the mechanisms by which the DE impacts carbon pollution through economic growth, industrial structure supererogation, financial development, and green innovation, there has been limited investigation into the ways in which the DE influences carbon pollution in terms of energy efficiency and foreign investment. Lastly, existing research has not adequately addressed the diverseness of the impact of the DE on carbon pollution from various perspectives, such as regions, city sizes, city resource endowment, and urbanization level. These limitations highlight the need for future research to address these gaps in knowledge and provide a more comprehensive understanding of the relationship between the DE and carbon pollution.

The purpose of this paper is to examine the impact of the DE on carbon pollution from both theoretical and empirical perspectives. The focus is on investigating the impact effect, action mechanism, and heterogeneity of the DE on carbon pollution. To achieve these research objectives, several steps are taken, as follows. First, the relationship and mechanism between the DE and carbon pollution are explored from a theoretical level. Second, the DID model is utilized to measure the policy effect of the DE on urban carbon pollution, and a series of robustness testing methods are conducted to validate the robustness of the research conclusions of the benchmark regression model. Third, in order to examine the mediating action mechanism of the impact of the DE on carbon pollution, the study employs the mediating effect model, taking into account two mechanism variables: energy efficiency and foreign investment. Finally, the study investigates the heterogeneity of the impact of the DE on carbon pollution by grouping regression analysis from various perspectives, including regions, city sizes, city resource endowments, and levels of urbanization.

The research presented holds significant implications, as it can offer crucial support for accurately assessing the carbon pollution reduction effects of the DE and informing relevant policy designs.

## 2. Theoretical Analysis of the Relationship between the DE and Carbon Pollution

### 2.1. The DE and Carbon Pollution

Sustainable development refers to the simultaneous consideration of economic, social, and ecological development. True sustainable development can only be achieved by simultaneously meeting economic sustainable development, social sustainable development, and ecological sustainable development. The DE utilizes digitized, networked, intelligent, and innovative technologies to impact the real economy and social development [12]. By integrating the DE into the real economy, it contributes to high-quality economic and social development while also improving environmental pollution and urban air quality. Therefore, the DE serves as a crucial driver for both sustainable economic and social development, as well as ecological sustainability. In terms of the borderless organization theory, the growth of the DE empowers the expeditious crossing of information, resources, creativity, and energy between corporations, permitting managers to respond blazingly to environmental variations [13]. This not only optimizes processing technology to achieve emission reduction from enterprises but also realizes a more energy-efficient production process to reduce carbon pollution [14]. From the perspective of environmental supervision, the digital economy can utilize information technology to achieve real-time monitoring of various pollutants and pollution sources. In particular, smart monitoring systems play a crucial role in encouraging polluting firms and industries to effectively control and eliminate excessive pollutant emissions, thereby achieving environmentally friendly production and reducing carbon pollution [15]. From the perspective of macro-governance, the government can leverage digital technology to acquire a better comprehension of energy trade direction and price oscillations. By regulating energy prices and employing cross-subsidies, the government can constrain the total energy provision and conduct the provision of clean energy. This enables the optimization of the energy supply system and reduces carbon pollution [16]. Additionally, the DE has revolutionized the way in which people work and live, especially during the pandemic. It has reduced the need for daily travel and transportation, promoted the adoption of home online offices, and efficiently decreased carbon pollution in work and life. As such, the following hypothesis is put forward.

**Hypothesis 1.** *The DE can have a significant influence on reducing carbon pollution.*

### 2.2. The Underlying Mechanism of the DE Acting on Carbon Pollution

Among the various factors contributing to the continuous rise in total carbon pollution, excessive reliance on fossil fuels is considered a significant cause. It is widely acknowledged that adopting a reasonable energy structure and improving energy efficiency are crucial strategies for reducing $CO_2$ emissions. The emergence of the DE has introduced fresh tools for managers to monitor the dynamics among producers, suppliers, and consumers, thereby facilitating the adaptation of production plans and achieving effective resource distribution. Simultaneously, the DE also disrupts conventional models of resource utilization. Various businesses have different preferences when it comes to implementing digital technologies, which can enhance resource allocation and boost energy efficiency [17]. Advancements in emerging digital technologies have led to increased automation, interconnectivity, and flexibility in the processes of production, manufacturing, and consumption [18,19]. These technologies also enable intelligent administration and monitoring of energy systems and data, which can be beneficial for improving energy efficacy. The DE heavily relies on information technology, which involves the input of knowledge, technology, and other elements. This approach presents superiorities such as low cost of diffusion, high marginal revenue, and scale efficiency. The integration of these novel elements with traditional industries can accelerate the optimization of product mix, improve energy efficacy, and facilitate the transformation towards low energy consumption and low carbon pollution. Ultimately, these initiatives will contribute to reducing pollution emissions. Additionally, the development of the DE facilitates the transformation from high-energy and high-pollution industries to low-energy, low-pollution, and knowledge-intensive industries [20]. This transition

towards industries with lower energy consumption and pollution levels is primarily attributed to the optimization and upgrading of the industrial structure enabled by the DE. Consequently, the growth of these low-energy-consumption and low-pollution industries has bolstered energy efficiency and indirectly mitigated carbon pollution. Therefore, the following hypothesis is proposed:

**Hypothesis 2.** *The DE can enhance energy efficiency, thereby reducing carbon pollution.*

The growth of the DE has the potential to enhance the market economy system, ameliorate information symmetry among market participants, eliminate information barriers between regions, ameliorate the efficiency of information transmission, decrease transaction expenditure for foreign investment, and ultimately attract more foreign investment [21]. While some scholars argue that developed countries have been transferring traditional industries with low added value and high pollution to China, which has contributed to the industrialization in China to some extent [22], this practice is not beneficial for the high-level development of the service industry or carbon pollution reduction. However, the Chinese economy has now reached a state of high-quality development. Emerging technologies, including blockchain, big data, artificial intelligence, and cloud computing, have played a remarkable part in making these information mechanisms in the economy and society more transparent [23]. These technologies have also alleviated the problem of information asymmetry to a great extent, reducing the flow barriers of factors in the market and decreasing the additional cost associated with foreign investment. Consequently, the DE has been successful in attracting foreign investment. Additionally, the DE has ameliorated the efficacy of foreign investment supervision and made it easier to guide foreign investors towards investing in the fields of energy conservation and environmental protection. This positive influence has facilitated the transformation of polluting industries, ultimately resulting in a significant diminution of carbon pollution. Therefore, it is hypothesized that the DE has the potential to attenuate carbon pollution with its ability to attract foreign investment.

**Hypothesis 3.** *The DE can contribute to the reduction of carbon pollution by attracting foreign investment.*

### 3. Research Design
#### 3.1. Model Setting

To assess whether the DE can contribute to reducing urban carbon pollution in China, we use the pilot of the National Big Data Comprehensive Pilot Zone as a case study to test the research hypothesis by evaluating the effectiveness of its policies. To achieve this, a two-way fixed-effect Difference-in-Differences (DID) model is constructed, taking advantage of the quasi-natural experiment provided by the pilot zone. The baseline regression model is specified as follows:

$$\ln CE_{it} = \beta_0 + \beta_1 did_{it} + \beta_2 Control_{it} + \eta_i + \delta_t + \varepsilon_{it} \tag{1}$$

In the formula provided, the subscripts $i$ and $t$ severally denote a city and a year. The $\ln CE_{it}$ represents the logarithm of city $i$'s per capita carbon dioxide pollution in year $t$, $did_{it}$ indicates if city $i$ implemented the National Big Data Comprehensive Pilot Zone pilot policy in year $t$. $Control_{it}$ denotes a set of control variables that influence urban carbon pollution and vary across cities and time. $\eta_i$ represents the individual fixed effect of the city that does not change over time. $\delta_t$ denotes the year fixed effect, which controls time-related factors that affect all cities that change over time. $\varepsilon_{it}$ is a residual term. The estimation coefficient $\beta_1$ is a policy effect concerned in this paper. Assuming that $\beta_1$ is significantly negative, this indicates that this policy is effective, meaning that the DE has a considerable inhibition effect on carbon pollution.

Based on the baseline regression about the identification of policy effects, we aim to probe if the DE impacts carbon pollution through two pathways: energy efficiency and foreign investment. To achieve this, we employ the mediating effect test approach put forward by Wen Zhonglin et al. (2014) and Lin Feng et al. (2022) to construct the following test equation [24,25]:

$$M_{it} = \alpha_0 + \alpha_1 \text{did}_{it} + \alpha_2 \text{Control}_{it} + \eta_i + \delta_t + \varepsilon_{it} \tag{2}$$

$$\ln CE_{it} = \varphi_0 + \varphi_1 \text{did}_{it} + \varphi_2 M_{it} + \varphi_3 \text{Control}_{it} + \eta_i + \delta_t + \varepsilon_{it} \tag{3}$$

wherein $M_{it}$ is a mechanism variable, representing energy efficiency (lnene) and foreign investment (lnfdi), and other variables have been explained earlier. The validity of the mediating effect will be determined by whether both coefficients $\alpha_1$ and $\varphi_2$ are significant. The Bootstrap method will be used to directly test the hypothesis $\alpha_1 \times \varphi_2 = 0$ if at least one of the coefficients is not significant. If $\alpha_1 \times \varphi_2$ significantly deviates from 0, it confirms the existence of a mediating effect. Conversely, if it does not significantly deviate from 0, the mediating effect is considered invalid. In the case of a mediating effect, it suggests that the direct effect is not significant, indicating the existence of only the mediating effect when the coefficient $\varphi_1$ is not significant. Conversely, if the coefficient $\varphi_1$ is significant, the direct effect is deemed significant, and it is necessary to compare the symbols of $\alpha_1 \times \varphi_2$ and $\varphi_1$. If the symbols are the same, it is considered a partial mediating effect, but if they are different, it's referred to as a masking effect.

### 3.2. Various Variable Settings

### 3.2.1. The Explained Variables

The explained variable in this study is carbon pollution, measured by per capita carbon dioxide emissions (lnCE). According to Cong et al. (2014), total carbon dioxide emissions can be measured across various sectors, including urban transport, construction, industrial production processes, land-use changes in agriculture and forestry, and greenhouse gas emissions from waste treatment activities. Furthermore, these measurements also take into account emissions from electricity, heating, or refrigeration required for urban consumption, as well as greenhouse gas emissions from the production, transport, use, and waste treatment of all goods purchased by cities from outside their jurisdiction [26]. The total carbon dioxide emissions of a city are calculated as the aggregate carbon pollution within the above scope. Meanwhile, the per capita carbon dioxide emissions are determined by dividing the total carbon dioxide emissions of the city by its permanent population.

### 3.2.2. Core Explanatory Variables

The National Big Data Comprehensive Pilot Zone is considered the core explanatory variable, denoted by the variable did. The calculation for did is $\text{did}_{it} = \text{treat}_i \times \text{post}_t$, where $\text{treat}_i$ denotes a dummy variable indicating whether city $i$ is attached to the treatment group, and $\text{post}_t$ also denotes a dummy variable representing the time of policy implementation. In 2016, 10 provinces and cities in China were authorized to establish National Big Data Comprehensive Experimental Zones. Hence, if the time $t \geq 2016$, $\text{post}_t = 1$, otherwise $\text{post}_t = 0$. If the city $i$ builds a National Big Data Comprehensive Pilot Zone, $\text{treat}_i = 1$, and if not, $\text{treat}_i = 0$.

### 3.2.3. Control Variables

Urban carbon pollution is also affected by many other factors. It is necessary to include the following control variables that have been identified in existing studies:

- The economic development (lngdp). This variable is gauged by per capita GDP, which is a crucial factor to consider. However, it is also important to take into account the theory of the environmental Kuznets curve [27], which suggests that an inverted U-shaped connection exists between economic development level and environmental

pollution. Therefore, the regression model includes the square term (lngdp2) of per capita GDP.

- Population density (lnpop). This variable is included in the study due to its significance in the IPAT model's understanding of urban carbon pollution [28,29]. On the one hand, higher population density usually leads to increased resource consumption, resulting in higher carbon pollution. On the other hand, the concentration of the population can also promote energy efficiency and the sharing of infrastructure, which in turn reduces carbon pollution [30]. Therefore, this paper incorporates population density as a control variable, measured by the ratio of the total population to the administrative area.

- Trade openness (lntrd). This variable plays a significant role in influencing carbon pollution through its impact on domestic technological progress and the transformation and upgrading of industrial structures. This is achieved through cross-border technological exchanges and the enforcement of stricter environmental access standards. As a result, trade openness is included as a control variable in this study, utilizing the total import and export volume of the city as a measure.

- Urbanization (lnurc). Existing studies have shown a strong relationship between urban carbon pollution and urbanization. On the one hand, the rapid advancement of urbanization leads to a substantial increase in energy consumption and, subsequently, carbon pollution. On the other hand, the progress made in urbanization facilitates the more concentrated use of urban energy, the optimization of energy structures, and the upgrading of industrial structures, all of which contribute to decreasing urban carbon pollution. Consequently, urbanization is considered a control variable in this study, measured by the ratio of the urban population to the total city population.

- Investment in fixed assets (lnfai). Investment in fixed assets plays a crucial role in influencing urban carbon pollution. Increased investment in urban fixed assets, particularly in environmental protection, not only promotes the development of environmental protection technologies but also facilitates the upgrading of industrial structures. Consequently, this ultimately helps to decrease urban carbon pollution. Thus, fixed asset investment is included as a control variable in this study, measured by the total amount of urban fixed asset investment.

- Social consumption (lnsoc). This variable is closely connected to urban carbon pollution. Social consumption, which is one of the three major drivers of economic growth, has significant implications for resource consumption and environmental issues. Traditional social consumption tends to increase resource consumption and contribute to environmental problems [31]. However, with an increased awareness among residents about environmental conservation, green consumption that promotes nature preservation and ecological protection can incentivize enterprises to produce more environmentally friendly products. As a result, urban carbon pollution can be reduced. Given the importance of this variable, social consumption is also considered a control variable in this study, measured by the ratio of total retail sales of urban social consumer goods to GDP.

### 3.2.4. Mechanism Variables

In light of the theoretical analysis of the action mechanism, this study selects two mechanism variables. One variable is energy efficiency (lnene), which denotes the output value per unit of energy consumption. It is denoted as the ratio of the city's GDP to its total energy consumption. Considering the possible lack of various energy consumption data for cities, the total energy consumption is converted into standard coal instead of the electricity consumption of the entire city. The other variable is foreign investment (lnfdi), which serves as a proxy variable for the real utilization of foreign investment.

*3.3. Data Origins and Processing*

Considering data accessibility, the balance panel data of 282 prefecture-level cities in China spanning from 2012 to 2021 were selected as a research sample. The data used primarily comes from sources such as statistical yearbooks of different levels in China, the Strawberry Scientific Research Service Network, Prospective Database, etc. In cases where data was missing, it was supplemented through linear interpolation. To reduce data discretization and mitigate the adverse effects of heteroscedasticity on equation estimation, logarithmic processing was applied to variables other than the core explanatory variables. The descriptive statistics of the variables are presented in Table 1.

**Table 1.** Descriptive statistics of the variables.

| Variable Class | Name | Sign | N | Mean | SD | Min | Max |
|---|---|---|---|---|---|---|---|
| Explained variables | Carbon pollution | lnCE | 2820 | 2.198 | 0.707 | 0.105 | 5.037 |
| Explanatory variables | DE | did | 2820 | 0.121 | 0.327 | 0.000 | 1.000 |
| | Economic development | lngdp | 2820 | 10.790 | 0.530 | 8.783 | 12.320 |
| | The square of economic development | $lngdp^2$ | 2820 | 116.800 | 11.470 | 77.150 | 151.900 |
| | Population density | lnpop | 2820 | 5.759 | 0.921 | 1.628 | 8.058 |
| | Trade openness | lntrd | 2820 | 14.020 | 2.110 | 3.211 | 21.330 |
| | Urbanization | lnurc | 2820 | 3.997 | 0.267 | 3.105 | 4.605 |
| | Investment in fixed assets | lnfai | 2820 | 7.192 | 0.915 | 4.086 | 9.992 |
| | Social consumption | lnsoc | 2820 | 3.614 | 0.359 | −5.773 | 4.618 |
| Mechanism variables | Energy efficiency | lnene | 2820 | 2.774 | 0.774 | −0.567 | 5.596 |
| | Foreign investment | lnfdi | 2820 | 11.78 | 2.105 | −5.015 | 16.83 |

Note: It should be noted that the variables, except for did, were logarithmically transformed for statistical analysis.

## 4. Empirical Results and Analysis

*4.1. Baseline Regression*

On the basis of the panel data from 282 of China's prefecture-level cities spanning from 2012 to 2021, we utilized a two-way fixed effect regression model to test Hypothesis 1 as specified in Equation (1). To avoid potential autocorrelation problems associated with the panel data, cluster-robust standard errors were used to estimate the standard errors. Table 2 presents the baseline regression consequences for the impact of the DE on carbon pollution. Column (1) displays the regression consequences without control variables, while columns (2) to (7) progressively include different control variables. The baseline regression consequences indicate that did has a consistently significant negative effect on urban carbon pollution, regardless of the inclusion of control variables. This suggests that the growth of the DE can effectively decrease carbon pollution in urban areas, thus providing support for research Hypothesis 1.

Considering the regression consequences of control variables in column (7), the primary coefficient of per capita GDP is significantly positive, while the secondary coefficient is significantly negative. This suggests that there are inverted "U"-type connections between economic development level and carbon pollution, which proves the existence of the environmental Kuznets curve. Additionally, population density has a considerable inhibitory impact on carbon pollution, indicating that the agglomeration effect of population density is dominant. The positive externality of this effect can reduce carbon pollution through cost-saving measures and technology spillover [32]. Furthermore, foreign trade has a noticeable inhibitory effect on carbon pollution, indicating that transnational technological exchanges and cooperation facilitated by foreign trade can promote technological progress and industrial structure upgrading in cities, which in turn can slow down urban carbon pollution. The coefficient of urbanization is significantly positive, indicating that the increase in urbanization level will lead to a rise in carbon pollution because of the expanded energy consumption demand. The coefficient between fixed asset investment

and carbon pollution is significantly negative. This finding is in accordance with the research consequences of Chen (2022) [33]. This suggests that fixed asset investment activities can result in production-scale effects and reduce carbon pollution. Social consumption has a significant positive influence on carbon pollution, demonstrating that an increase in social consumption will stimulate the supply level of enterprises, resulting in higher energy consumption and increased carbon pollution.

**Table 2.** Test results of the DE on carbon pollution.

| Variables | (1) | (2) | (3) | (4) | (5) | (6) | (7) |
|---|---|---|---|---|---|---|---|
| did | −0.043 *** | −0.038 *** | −0.022 ** | −0.022 * | −0.026 ** | −0.021 * | −0.021 * |
| | (0.012) | (0.012) | (0.011) | (0.011) | (0.011) | (0.012) | (0.012) |
| lngdp | | 2.335 *** | 2.059 *** | 2.098 *** | 1.935 *** | 1.924 *** | 1.871 *** |
| | | (0.573) | (0.579) | (0.572) | (0.634) | (0.576) | (0.568) |
| $lngdp^2$ | | −0.104 *** | −0.090 *** | −0.091 *** | −0.084 *** | −0.080 *** | −0.078 *** |
| | | (0.026) | (0.026) | (0.025) | (0.028) | (0.025) | (0.025) |
| lnpop | | | −0.464 *** | −0.455 *** | −0.453 *** | −0.399 *** | −0.400 *** |
| | | | (0.117) | (0.114) | (0.112) | (0.112) | (0.112) |
| lntrd | | | | −0.019 ** | −0.020 ** | −0.015 ** | −0.016 ** |
| | | | | (0.008) | (0.008) | (0.008) | (0.008) |
| lnurc | | | | | 0.103 * | 0.088 | 0.096 * |
| | | | | | (0.058) | (0.055) | (0.055) |
| lnfai | | | | | | −0.065 *** | −0.065 *** |
| | | | | | | (0.018) | (0.018) |
| lnsoc | | | | | | | 0.027 * |
| | | | | | | | (0.016) |
| Constant | 2.203 *** | −10.857 *** | −6.835 ** | −6.892 ** | −6.424 * | −6.575 ** | −6.419 ** |
| | (0.002) | (3.222) | (3.434) | (3.371) | (3.574) | (3.292) | (3.255) |
| Urban fixed effect | Yes | Yes | Yes | Yes | Yes | Yes | Yes |
| Year fixed effect | Yes | Yes | Yes | Yes | Yes | Yes | Yes |
| Sample size | 2820 | 2820 | 2820 | 2820 | 2820 | 2820 | 2820 |
| Adj. $R^2$ | 0.976 | 0.978 | 0.978 | 0.979 | 0.979 | 0.979 | 0.979 |

Note: "***", "**", and "*" express significance levels at 1%, 5%, and 10%, respectively. The numbers in brackets are cluster-robust standard errors.

*4.2. Parallel Trend Test*

The primary premise of implementing the DID model is that both the treatment group and the control group should exhibit parallel trend hypotheses. In other words, when the experimental group is not subject to policy intervention, its time trend effect should theoretically be consistent with that of the control group [34]. Currently, there are two methods for testing parallel trend tests: the time trend chart and the event research method. The time trend chart, illustrated in Figure 1, demonstrates that the carbon pollution (lnCE) of the pilot in the treatment group and the control group is not systematically different over the time before 2015, indicating a parallel trend. Therefore, Figure 1 clearly illustrates that this study satisfies the prerequisite for utilizing the DID model.

Compared to the time trend chart, the event research method is considered to be more accurate because the time trend chart relies on only subjective determination to identify whether there exists a pronounced divide. Within the study, this event research method is employed to conduct a parallel trend test using 2012 as the reference period. The results of this method are presented in Figure 2, wherein the horizontal axis denotes the year, the vertical axis denotes the estimated effect value of the processing effect, the dots in the figure represent the policy effect of the treatment group established in the National Big Data Comprehensive Pilot Zone in the year, and the dashed line represents the 10% confidence interval. It is evident from the figure that there was no statistically significant difference in the carbon pollution reduction effect between the treatment group and the control group before the establishment of the National Big Data Comprehensive Pilot Zone.

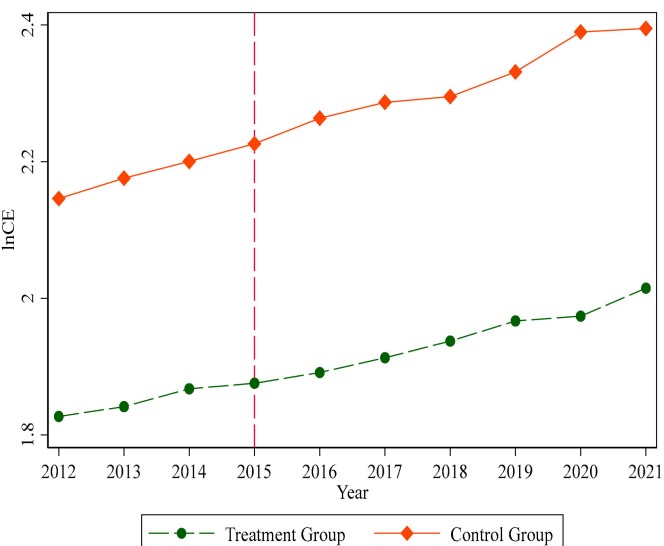

**Figure 1.** Time trend chart for lnCE change.

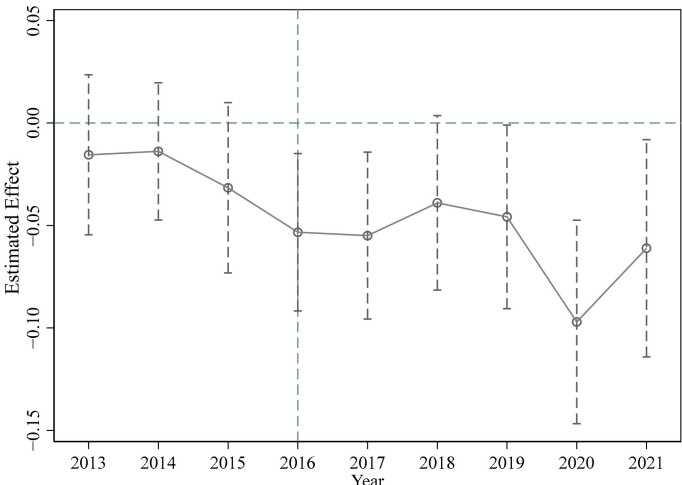

**Figure 2.** The event research method for parallel trends.

In the year following the establishment of the National Big Data Comprehensive Pilot Zone, the carbon pollution reduction effect of the treatment group, with the exception of 2018, is significantly reduced compared to the control group. This suggests that the DID model adopted in this research meets the parallel trend assumption.

*4.3. Placebo Test*

To examine whether the regression results in Table 2 are influenced by missing variables and random factors, a permutation test is employed to conduct a placebo test [35]. This involves randomly sampling all cities and policy time, selecting 57 cities each time, and generating a random time. The 57 cities selected each time are considered the dummy treatment group, while the remaining cities serve as the dummy control group. This process is repeated 500 times to obtain the regression coefficients of the interaction terms of the 500 dummy treatment groups and their policy time. The coefficient distribution graph presented in Figure 3 indicates that the coefficients of the spurious did terms are centered around zero, and the majority of the corresponding *p*-values are bigger than 0.1. These results are consistent with the normal distribution. Moreover, in column (7) of Table 2, the coefficient did for the interaction term is $-0.021$, which is located at the low end of the distribution of the coefficient obtained through the permutation test. The placebo test indicates that there are no significant problems with missing variables or random factors

interfering in the model setting. As a result, the conclusions of the previous study are considered to be relatively robust.

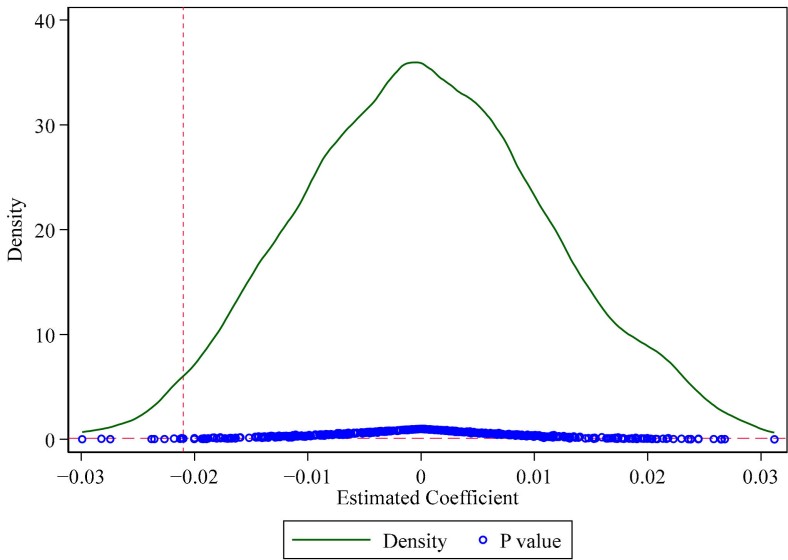

**Figure 3.** Placebo tests.

*4.4. Other Robustness Tests*

4.4.1. PSM–DID Estimation

In order to avoid the endogeneity issue due to the potential non-random selection of policies, we refer to the methods employed by Xu Si et al. (2019) and Wang et al. (2023) [36,37]. We further utilize a combination of propensity score matching (PSM) and the DID model to regress the model again. Firstly, the dummy variable of whether the city is a National Big Data Comprehensive Pilot Zone is taken as the explained variable, and all control variables in the baseline regression model are taken as the explanatory variables. The logit regression is carried out, and the corresponding propensity score value for each sample is calculated. Secondly, according to the propensity score, the nearest neighbor matching is carried out among the cities that are not yet in the National Big Data Comprehensive Pilot Zone, and a group of cities with a similar propensity score to the cities in the National Big Data Comprehensive Pilot Zone is picked out as the control group.

Table 3 illustrates that after matching, the deviation between pilot cities and non-pilot cities is significantly reduced, the mean value of covariates is not significantly different, and the standard deviation absolute value is controlled below 10%, suggesting that the matching effect is ideal. The effectiveness of the matching can also be observed in Figure 4, which displays the kernel density maps of propensity scores both before and after matching. From the figure, the deviation of the former two kernel density curves is relatively large when matched, but the latter two curves are close to each other, and the distance between the mean lines is shortened after matching. Therefore, it further indicates that PSM has a good matching effect by reducing sample selectivity bias. Lastly, the regression analysis is conducted again using the matched samples described above, and the specific results are presented in column (1) of Table 4. It is found that the research conclusion of this paper remains valid even after applying PSM in the regression analysis. Additionally, other matching methods such as kernel matching, radius calipers matching, and Mahalanobis distance matching are also employed, and the regression results are consistently aligned with the aforementioned conclusions.

**Table 3.** Balance testing results.

| Variables | Unmatched | Mean Value | | Bias (%) | Reduct | t-Test | |
|---|---|---|---|---|---|---|---|
| | Matched | Treated | Control | | \|Bias\| (%) | t | p > \|t\| |
| lngdp | U | 10.890 | 10.768 | 23.000 | 77.800 | 4.930 | 0.000 |
| | M | 10.890 | 10.917 | −5.100 | | −0.860 | 0.388 |
| lngdp$^2$ | U | 118.870 | 116.220 | 23.000 | 78.000 | 4.950 | 0.000 |
| | M | 118.870 | 119.460 | −5.100 | | −0.850 | 0.396 |
| lnpop | U | 5.999 | 5.698 | 30.000 | 85.500 | 7.010 | 0.000 |
| | M | 5.999 | 5.955 | 4.300 | | 0.750 | 0.452 |
| lntrd | U | 14.550 | 13.891 | 30.300 | 98.900 | 6.720 | 0.000 |
| | M | 14.550 | 14.543 | 0.300 | | 0.060 | 0.954 |
| lnurc | U | 4.044 | 3.985 | 20.800 | 90.100 | 4.720 | 0.000 |
| | M | 4.044 | 4.050 | −2.100 | | −0.360 | 0.718 |
| lnfai | U | 7.620 | 7.084 | 62.200 | 93.500 | 12.850 | 0.000 |
| | M | 7.620 | 7.585 | 4.000 | | 0.710 | 0.480 |
| lnsoc | U | 3.581 | 3.623 | −12.800 | 29.400 | −2.510 | 0.012 |
| | M | 3.581 | 3.551 | 9.000 | | 1.400 | 0.161 |

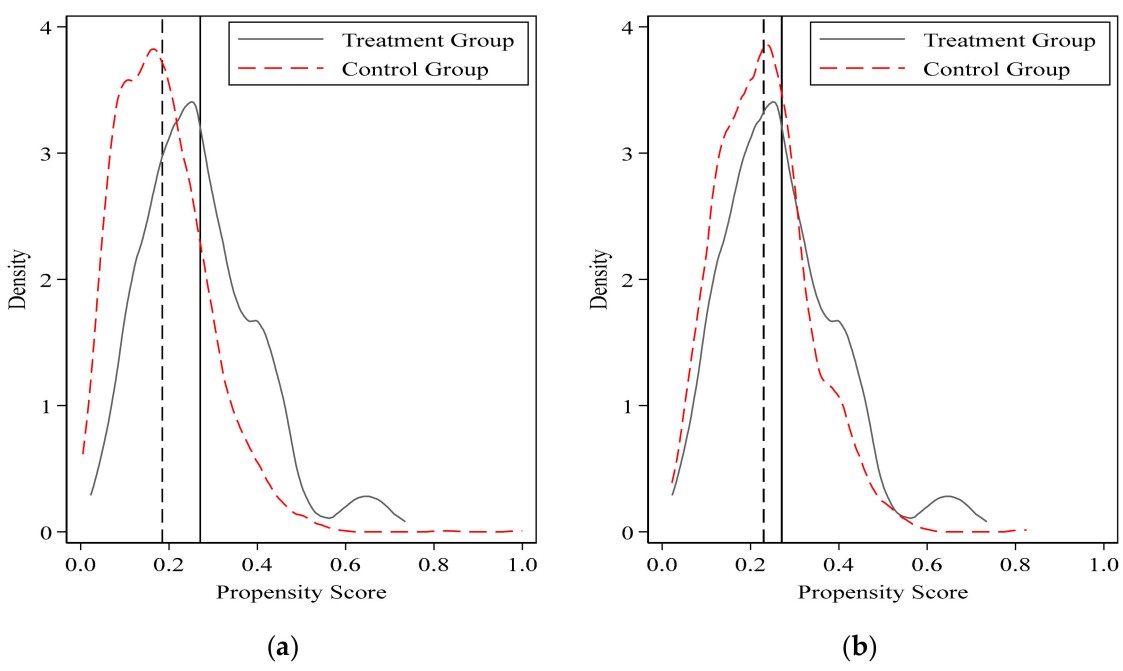

**Figure 4.** Propensity score probability distribution density function. (**a**) Propensity score before matching; (**b**) propensity score after matching.

### 4.4.2. Excluding the Influence of Contemporaneous Relevant Policies

Policies related to the DE implemented during the sample period may also influence the carbon pollution of the pilot cities, which might potentially interfere with the estimation of the carbon pollution reduction effects of pilot policies in the National Big Data Comprehensive Pilot Zone. Besides the pilot cities in the National Big Data Comprehensive Pilot Zone, two other initiatives, the "Smart City" pilot launched from 2012 to 2014 and the "Broadband China" initiative implemented from 2014 to 2016, are closely linked to this study [38,39]. Inspired by the methodologies employed by Bai Junhong et al. (2022) [16] and Zhi Yupeng et al. (2023) [40], this paper incorporates dummy variables for the "Smart City" and "Broadband China" pilot policies into the baseline regression model in a consec-

utive manner, aiming to isolate the impacts of relevant policies. The values presented in columns (2) and (3) of Table 4 indicate that the did coefficient remains significantly negative. Furthermore, when considering the policy coupling factor, both the pilot policies of "Smart City" and "Broadband China" are included in the model simultaneously, as shown in column (4) in Table 4. In this case, the coefficient of the pilot policy variables in the National Big Data Comprehensive Pilot Zone remains significantly negative at the 10% significance level. These findings suggest that the pilot policies of "Smart city" and "Broadband China" have only a limited influence on the carbon pollution reduction effect of the pilot policies in the National Big Data Pilot Zone.

**Table 4.** Results of other robustness tests.

| Variables | (1) PSM-DID | (2) Policies Excluded Related to the Same Period | (3) | (4) | (5) Transforming the Explained Variable | (6) | (7) Excluding Samples | (8) Winsorize | (9) Transformation Explanatory Variable |
|---|---|---|---|---|---|---|---|---|---|
| did | −0.027 ** (0.013) | −0.020 * (0.012) | −0.021 * (0.012) | −0.020 * (0.012) | −0.016 ** (0.008) | −0.021 * (0.012) | −0.024 * (0.013) | −0.026 **(0.012) | |
| $did_1$ | | −0.017 (0.013) | | −0.016 (0.012) | | | | | |
| $did_2$ | | | −0.014 (0.011) | −0.013 (0.011) | | | | | |
| dig | | | | | | | | | −0.028 * (0.016) |
| Constant | −6.489 *** (2.384) | −6.333 * (3.248) | −6.218 * (3.331) | −6.145 * (3.322) | 7.649 *** (0.994) | 2.791 (3.255) | −5.517 (3.647) | −5.490 ** (2.402) | −6.748 *** (1.152) |
| Control variables | Yes | Yes | Yes | Yes | Yes | Yes | Yes | Yes | Yes |
| Urban fixed effect | Yes | Yes | Yes | Yes | Yes | Yes | Yes | Yes | Yes |
| Year fixed effect | Yes | Yes | Yes | Yes | Yes | Yes | Yes | Yes | Yes |
| Sample size | 1510 | 2820 | 2820 | 2820 | 2820 | 2820 | 2470 | 2820 | 2820 |
| Adj. $R^2$ | 0.978 | 0.979 | 0.979 | 0.979 | 0.977 | 0.986 | 0.977 | 0.979 | 0.979 |

Note: Symbols $did_1$ and $did_2$ are policy dummy variables for "smart city" and "Broadband China", respectively. Others are the same as before. "***", "**", and "*" express significance levels at 1%, 5%, and 10%, respectively. The numbers in brackets are cluster-robust standard errors.

### 4.4.3. Transforming the Explained Variable

The baseline regression model employs per capita carbon dioxide emissions as a measure of carbon pollution levels in cities. To strengthen the robustness of the research findings, two alternative indicators, namely total carbon emission and carbon emission intensity (the ratio of the total carbon emission of the city to the city's gross domestic product), are employed as measurement indicators for the explained variables. The regression results for the explained variables of total carbon emission and carbon intensity are presented in columns (5) and (6) of Table 4, respectively. It can be observed that even after substituting the measurement indicators for the explained variables, the coefficient of the did coefficient remains significantly negative, consistent with the results of the baseline regression.

### 4.4.4. Sample Elimination and Variable Winsorize

To eliminate the possible intervention of urban development imbalance on the research conclusion, it is acknowledged that there may be significant differences between municipalities, provincial capitals, sub-provincial cities, and general prefecture-level cities in carbon pollution, the DE, and economic development. Hence, samples from municipalities, provincial capitals, and sub-provincial cities are excluded, and only samples from general cities are retained. The regression consequences are presented in column (7) of Table 4. The evaluated coefficient of the did remains significantly negative. This indicates that even after removing samples of cities with high administrative ranks, the research

conclusion of this paper remains stable. Furthermore, to mitigate the effect of outliers on the consequences, both the explained variables and all control variables were subjected to 1% winsorization. The regression results after winsorization for each tail are presented in column (8) of Table 4. The evaluated coefficient for the pilot policy variables in the National Big Data Comprehensive Pilot Zone remains significantly negative. This highlights the robustness of the baseline regression results even after eliminating those variable outliers.

### 4.4.5. Transformation Variables

In order to obtain a more comprehensive understanding of the relationship between the level of DE development and urban carbon pollution, the core explanatory variable of this paper, which is a dummy variable indicating whether a city has established a National Big Data Comprehensive Pilot Zone, needs to be transformed. The regression results, based solely on this dummy variable, may not fully capture the relationship between DE development and urban carbon pollution. To address this limitation, a measurement index system for the DE is adopted, following the approach of Zhao Tao, Zhang Zhi, and Liang Shangkun (2020) [12].

The measurement index system includes variables such as the Internet penetration rate, Internet-related output, the number of mobile Internet users, the inclusive finance index, and the number of employees in the Internet-related sector. These variables are used to construct the DE Development Index (dig) using the entropy method. In the regression analysis, the dummy variable (did) is replaced with the DE Development Index (dig), and the results are presented in column (9) of Table 4. The coefficient of the dig variable is −0.028, and it passes the significance test at the 10% level. This finding further strengthens the research conclusion of this paper.

### 4.5. Analysis of Action Mechanism

The above baseline regression results indicate the DE has a significant influence on reducing the carbon pollution of pilot cities, and the research findings are robust. However, the specific mechanism of this impact is not yet clear. The theoretical analysis indicates that the DE may decrease carbon pollution through two channels: ameliorating energy efficacy and attracting foreign investment. The test results of the action mechanism, as given in Table 5, are derived from Equations (2) and (3). Columns (1) and (2) in Table 5 display the test results for energy efficiency as the action mechanism variable. In column (1), the coefficient of the did is significantly positive, indicating that the pilot policy has a substantial positive effect on energy efficiency. In column (2), the coefficient of the did is significantly negative. The coefficient of energy efficiency, denoted as lnene, is significantly negative. The product of the coefficient did in column (1) and the coefficient lnene in column (2) has the same negative sign as the coefficient did in column (2). This suggests the presence of a relatively significant partial mediating effect. The mediating effect of energy efficiency accounts for nearly 13.47% of the gross reduction in carbon pollution attributed to the DE. This indicates that improving energy efficiency can help reduce carbon pollution in the DE. Therefore, research Hypothesis 2 has been verified, asserting that the DE can decrease carbon pollution through the enhancement of energy efficiency.

The test results for the action mechanism of foreign investment (lnfdi) are presented in columns (3) and (4) of Table 5. In column (3), the regression results show a significantly positive coefficient for the dummy variable did, indicating that the pilot policy of the National Big Data Comprehensive Pilot Zone has a significant impact on attracting foreign investment. Meanwhile, in column (4), the regression results reveal a significantly negative coefficient for the variable did. Moreover, the coefficient for foreign investment (lnfdi) is also significantly negative. The symbol of the product of the coefficients for did in column (3) and lnfdi in column (4) aligns with that of the coefficient for did in column (3). Both of them are negative, indicating that there is a significant partial intermediary effect. The analysis suggests that the intermediary effect of foreign investment accounts for approximately 11.54% of the total carbon pollution reduction effect of the DE. This finding

shows that the DE, through enhancing energy efficiency, can reduce carbon pollution by attracting foreign investment. It is important to note potential conflicts between the mechanism variable foreign investment (lnfdi) and the control variable trade openness (lntrd) in testing the intermediary action mechanism of foreign investment, which will affect the test results. However, the variance inflation factor (VIF) test for multicollinearity reveals that both of the VIF values for foreign investment (lnfdi) and trade openness (lntrd) are 2.40 and 2.97, respectively. They are below 5, indicating no issues of multicollinearity that could affect the test results for the action mechanism of foreign investment. Thus, the verification results for Hypothesis 3 of this study can be considered reliable.

**Table 5.** Test results of the action mechanism.

| Variables | (1) lnene | (2) lnCE | (3) lnfdi | (4) lnCE |
|---|---|---|---|---|
| did | 0.123 *** | −0.018 * | 0.303 *** | −0.018 * |
| | (0.037) | (0.010) | (0.106) | (0.010) |
| lnene | | −0.023 *** | | |
| | | (0.005) | | |
| lnfdi | | | | −0.008 *** |
| | | | | (0.002) |
| Constant | 51.741 *** | −5.255 *** | 7.822 | −6.355 *** |
| | (4.171) | (1.155) | (17.006) | (1.122) |
| Control variables | Yes | Yes | Yes | Yes |
| Urban fixed effect | Yes | Yes | Yes | Yes |
| Year fixed effect | Yes | Yes | Yes | Yes |
| Sample size | 2820 | 2820 | 2820 | 2820 |
| Adj. $R^2$ | 0.763 | 0.979 | 0.780 | 0.979 |

Note: "***", and "*" express significance levels at 1%, and 10%, respectively. The numbers in brackets are cluster-robust standard errors.

### *4.6. Further Analysis*

This paper aims to investigate the influence of the DE on carbon pollution and the underlying mechanism. It acknowledges that the impact of the DE on carbon pollution might vary across different regions, city sizes, resource endowments, and levels of urbanization.

### 4.6.1. Regional Heterogeneity

Regional heterogeneity refers to the variations or differences in certain aspects or characteristics among different regions. In this paper, it refers to the distinctions in economic development, industrialization stage, and resource endowment among cities, and how these differences may impact the connection between the DE and carbon pollution. To investigate this heterogeneity, the study divides the 282 prefecture-level cities in China into two categories: eastern cities and central and western cities [41]. This division allows for an exploration of how the DE affects carbon pollution in different regions. The regression results for regional heterogeneity are presented in columns (1) and (2) of Table 6. The findings indicate the coefficient for did in eastern cities is significantly negative, suggesting the DE has a significant negative effect on carbon pollution in the eastern districts. However, the coefficient for did in central and western cities is negative but not statistically significant, implying that the DE does not have a considerable effect on carbon pollution in the central and western regions. Due to the advanced economic growth in the eastern regions and the early establishment of digital infrastructure and industries, the level of DE growth is higher in comparison to the central and western regions. This has enabled the full realization of the carbon pollution reduction benefits associated with the DE. Conversely, the development of the DE in the central and western regions is still in its initial stages. These regions have relatively underdeveloped digital infrastructure and have taken over polluting industries in the east. Consequently, the DE in the central and western regions

has not been able to significantly reduce carbon pollution [42]. It is important to note that the regional heterogeneity analysis in this paper is similar to that of Li Y, Yang X, Ran Q et al. (2021) [43], but there are also significant differences. Li Y, Yang X, Ran Q et al. (2021) primarily focused on examining the regional heterogeneity in the regulatory effect of the DE on the relationship between energy structure and carbon pollution. In contrast, this paper concentrates on exploring the regional heterogeneity impact of the DE on carbon pollution.

**Table 6.** Results of heterogeneity tests.

| Variables | (1) East Cities | (2) Midwest Cities | (3) Mega Cities | (4) Big Cities | (5) Small and Medium Cities | (6) Resource-Based Cities | (7) Non-Resource-Based Cities | (8) High Urbanization Cities | (9) Low Urbanization Cities |
|---|---|---|---|---|---|---|---|---|---|
| did | −0.020 * | −0.027 | −0.001 | −0.037 ** | 0.153 | −0.013 | −0.023 * | −0.032 * | −0.010 |
| | (0.012) | (0.019) | (0.016) | (0.016) | (0.132) | (0.023) | (0.013) | (0.018) | (0.015) |
| Constant | −5.218 ** | −6.921 | −6.022 *** | −1.242 | −21.294 * | −8.494 | −4.287 ** | −12.084 * | −2.418 |
| | (2.498) | (4.384) | (2.121) | (2.422) | (11.822) | (6.027) | (2.049) | (6.331) | (1.972) |
| Control variables | Yes | Yes | Yes | Yes | Yes | Yes | Yes | Yes | Yes |
| Urban fixed effect | Yes | Yes | Yes | Yes | Yes | Yes | Yes | Yes | Yes |
| Year fixed effect | Yes | Yes | Yes | Yes | Yes | Yes | Yes | Yes | Yes |
| Sample size | 1000 | 1820 | 899 | 1830 | 84 | 1120 | 1700 | 1331 | 1489 |
| Adj. $R^2$ | 0.990 | 0.976 | 0.979 | 0.966 | 0.964 | 0.974 | 0.982 | 0.355 | 0.507 |

Note: "***", "**", and "*" express significance levels at 1%, 5%, and 10%, respectively. The numbers in brackets are cluster-robust standard errors.

### 4.6.2. Urban Size Heterogeneity

On the one hand, in comparison to small cities, large cities have higher resource allocation and utilization efficiency advantages [44]. These factors contribute to a diminution in environmental pollution. On the other hand, overcrowded cities tend to experience congestion effects, exacerbating urban diseases and environmental pollution problems [45]. Hence, it is essential to investigate whether the carbon pollution reduction effect of the DE is influenced by the heterogeneity of city sizes. In terms of the 2014 Notice on the Adjustment of Urban Size Division Standards released by the State Council, cities with a permanent population of over 5 million are classified as megacities, those with a population between 1 million and 5 million are classified as large cities, and those with a population below 1 million are classified as small and medium-sized cities [46]. The heterogeneity regression results for city size are presented in columns (3) to (5) of Table 6. The coefficient did for small and medium-sized cities is positive, while that for megacities is negative. However, both coefficients are not statistically significant. In addition, the coefficient for large cities is significantly negative. This indicates that the DE plays a significant role in reducing carbon pollution in large cities, but its impact is not significant in small and medium-sized cities, as well as megacities. One possible explanation is that larger cities, in comparison to small and medium-sized cities, possess superior digital infrastructure, DE-related technologies, capital, and a pool of high-quality talent. These advantages enable larger cities to foster technological advancements in the energy field through knowledge sharing and technology spillover [47]. As a result, larger cities experience a significant reduction in carbon pollution. However, too extensive a scale of cities also leads to increased energy consumption and congestion effects. Although the DE can curb carbon pollution, the effect is not obvious.

### 4.6.3. Heterogeneity in Urban Resource Endowment

Resource endowment has a significant impact on various aspects of urban technological progress and development patterns, including the DE. As a result, the effects of carbon reduction in DE may vary due to variations in resource endowments [48]. In terms of the National Sustainable Development Plan for Resource-Based Cities in China

(2013–2020) released by the State Council [49], the sample cities are divided into two categories: resource-based cities and non-resource-based cities. Columns (6) and (7) of Table 6 present the regression results for the impact of urban resource endowment heterogeneity on carbon pollution. The results demonstrate the DE has a considerable reduction effect on the carbon pollution of non-resource-based cities. However, it has a negative but statistically insignificant effect on the carbon pollution of resource-based cities. This suggests the DE has a greater potential for carbon pollution diminution in non-resource-based cities. The discrepancy can likely be attributed to the fact that resource-based cities have abundant fossil energy and mineral resources. Historically, these cities have relied heavily on these resources to develop resource-intensive industries, neglecting the development of environmental protection technologies. Consequently, carbon pollution has increased in these cities. Moreover, the level of digital industrialization in resource-intensive industries has been relatively low, which limits the extent to which the DE can be helpful to carbon pollution diminution in these cities.

4.6.4. Urbanization Heterogeneity

Due to the late start of China's urbanization and the rapid pace of development, the initial phase was primarily characterized by extensive expansion. Consequently, the industrial structure has developed in an unbalanced manner, hindering the realization of the carbon pollution diminution benefits of the DE. When urbanization reaches the middle and late stages, there are several benefits to reducing $CO_2$ emissions. First, the centralized utilization of resources, coupled with the spillover of information and technological progress brought about by urbanization, can enhance resource efficiency. This alternately leads to a reduction in carbon pollution. Second, as urbanization progresses, the growth of digital services reaches a new level. This can greatly facilitate the efficient flow of resources, minimizing resource consumption loss and ultimately mitigating urban carbon pollution [50]. Therefore, the impact of the DE on carbon pollution reduction may vary depending on the level of urbanization. Based on the median value of urbanization, urbanization values in the sample cities above the median are categorized as high-urbanization cities, while cities with urbanization values below the median are classified as low-urbanization cities. To evaluate the heterogeneity concerning the effect of urbanization on carbon pollution reduction, we present the regression results in columns (8) and (9) of Table 6. The findings indicate that, when compared to areas with low levels of urbanization, the DE significantly improves carbon pollution in highly urbanized regions.

## 5. Comparison with Existing Studies

Current literature studies primarily explore the DE's impact on economic development. For instance, Lu and Zhu (2022) found a significant spatial correlation between the DE and high-quality economic development, with the DE directly driving the latter through notable spillover effects [51]. Jiang and Deng (2022) similarly concluded that the DE has a substantial positive impact on high-quality economic development, displaying distinct non-linear characteristics [52]. However, beyond its influence on economic development, it is also crucial to explore whether the DE can offer novel solutions for carbon pollution reduction. Considering the correlation between high-quality economic development and environmental pollution, this paper aims to further investigate the impact of the DE on carbon pollution, thus expanding the scope of research on the DE in the environmental domain. The findings of this research align with the studies conducted by Li and Wang (2022) [53] and Yan et al. (2023) [54], both supporting the notion that the DE serves as an effective means of mitigating carbon pollution. In comparison to previous literature that employed index systems to measure the DE, this paper employs the National Big Data Comprehensive Pilot Zone. By utilizing the DID model, it comprehensively evaluates the net effect of the DE on carbon pollution, effectively addressing endogeneity concerns and testing the robustness of the analysis results, thus contributing to more reliable research conclusions.

## 6. Research Conclusions and Policy Recommendations

There is no doubt that the DE has become increasingly significant for high-quality economic development. With the dual goals, it is crucial to determine whether the thriving DE can effectively reduce carbon pollution while promoting economic growth. This paper aims to empirically investigate the connection between the DE and carbon pollution utilizing 282 of China's cities, spanning from 2012 to 2021. The findings of this study are outlined below:

- After conducting various robustness tests, including the parallel trend test, placebo test, PSM–DID estimation, exclusion of the influence of relevant policies, transformation of explained variables, removal of samples, and variable winsorize, it can be concluded that the DE has a considerable effect on reducing urban carbon pollution.
- The DE achieves this by enhancing energy efficiency and attracting foreign investment, which helps reduce carbon pollution effectively.
- The diminution in carbon pollution is more noticeable in the eastern regions compared with the central and western regions. Moreover, cities with excessively large or small scales do not experience a considerable diminution in carbon pollution through the DE. Instead, moderate-sized cities benefit the most from the carbon reduction effect. Additionally, non-resource-based cities and highly urbanized cities demonstrate a notable decrease in carbon pollution through the DE.

Based on the research findings mentioned above, this study proposes the following policy recommendations:

Firstly, on the basis of the establishment of the National Big Data Comprehensive Pilot Zone, a key initiative is to develop the DE, which can effectively contribute to reducing carbon pollution, thus playing a significant role in achieving environmental sustainability. The construction of the National Big Data Comprehensive Pilot Zone should be continued at a national level due to the significant reduction in carbon pollution resulting from the DE. It is important to consistently analyze the experience gained from the initial pilot zone and progressively develop a replicable and scalable construction approach. This pilot scope should be expanded based on the functional layout, geographical region, and urban characteristics of the pilot zone. This expansion should include a broader scope of application scenarios for the DE, with an emphasis on accelerating the development of emerging digital industrial clusters. It is crucial to ensure that the DE promotes economic development while effectively curbing carbon pollution. This can be achieved by simultaneously advancing economic growth and reducing carbon pollution, thus realizing a win-win situation. Overall, by implementing these policy recommendations, we can maximize the potential of the DE to spur economic growth while contributing to carbon pollution reduction.

Secondly, it is crucial to give due consideration to the mediating role of energy efficiency and foreign investment in reducing carbon pollution within the DE. On the one hand, efforts should be made to enhance connectivity in energy-related sectors, utilize more energy-efficient production equipment, foster the development of low-energy consumption products, and minimize energy consumption and losses throughout the usage process. On the other hand, seizing the opportunities presented by foreign investment in the DE era is essential. It is crucial to reform the traditional approach to attracting foreign investment and instead focus on strengthening investment in areas such as the Internet of Things, artificial intelligence, cybersecurity, blockchain, and global digital business.

Thirdly, it is imperative for cities to adopt tailored strategies for the development of the DE, considering the heterogeneity of regional, city size, urban resource endowments, and levels of urbanization.

- In the eastern region, it is important to not only focus on enhancing the application and innovation of digital technology but also take the lead in addressing the key technological challenges associated with reducing carbon pollution through the DE. Meanwhile, the central and western regions, as well as resource-based cities, should

accelerate the development of sustainable infrastructure projects. They can leverage their resource advantages to attract capital and talent in order to promote sustainable growth and reduce carbon pollution.

- It is necessary to rigorously control the scale of development in megacities, steer the development of large cities in a rational manner, and actively encourage the growth of small and medium-sized cities into larger ones. This will enable cities to fully capitalize on the advantages of economies of scale and agglomeration while reducing urban carbon pollution.

- It is important to promote a new form of urbanization. The current level of urbanization is generally not conducive to reducing carbon pollution; however, high-level urbanization can facilitate the carbon reduction potential of the DE. This suggests that urbanization has moved beyond the phase of extensive expansion. Therefore, it is crucial to promote the growth of new urbanization and strive for the construction of green, low-carbon, cyclic, ecological, and livable cities.

Of course, this paper also has some limitations. The analysis only focuses on the action mechanism of the DE on carbon pollution through the lenses of energy efficiency and foreign investment. It is worthwhile to explore additional mechanisms from other perspectives. Additionally, the regression analysis mainly relies on the classic DID model, which ignores the spatial correlation between individuals. Therefore, the research conclusions may not be fully comprehensive. Future studies should consider spatial correlation and investigate the effects of the DE on carbon pollution.

**Author Contributions:** Conceptualization, G.L.; Data curation, G.L.; Formal analysis, M.L.; Writing—original draft, M.L. and G.L.; Writing—review and editing, G.L. All authors have read and agreed to the published version of the manuscript.

**Funding:** This research was funded by the Social Science Planning Office, Foshan City, Guangdong Province (2023-GJ039).

**Institutional Review Board Statement:** Not applicable.

**Informed Consent Statement:** Not applicable.

**Data Availability Statement:** The data in this paper are from the statistical yearbook of Chinese cities and the statistical yearbook of prefecture-level cities.

**Conflicts of Interest:** The authors declare no conflict of interest.

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
