# Peer review of "Research on the Impact of the Digital Economy on Carbon Pollution Based on the National Big Data Comprehensive Pilot Zone in China"

_sustainability, doi:10.3390/su152115390_

Round 1

Reviewer 1 Report

It is an honour to review this paper. The paper is relatively well structured, with a sound methodology and reasonable conclusions. It is a relatively standardised and standard paper. However, I think it still has a lot of problems to be solved, especially the data of the paper, and I have some questions. Here are some suggestions: I hope some of my suggestions can help the author to revise the paper and improve its quality.

1.The presentation of formulas in the manuscript should be more standardised, and lines 174, 176 and 190 likewise require attention. For example: lnCO2.
2.The manuscript should briefly explain the reasons for the choice of control variables, is it based on a particular theoretical framework, or theoretical analysis? Rather than simply listing the control variables, in lines 233-242.
3. Minimum values for Social consumption and Energy efficiency, please confirm. I am particularly puzzled by the existence of negative values for efficiency. As you mentioned in the manuscript: One variable is energy efficiency (lnene) which is the output value per unit of energy consumption. 
4. Is there some conflict between lnfdi and lntrd? Could you please report the VIF test in the manuscript? I have some doubts about the overall results of the manuscript.
5. The study makes some assumptions, but does not respond to each of them in the manuscript, which I think makes the structure of the manuscript flawed.
6. Please reorganise the final section on policy recommendations. The policy recommendations also need to be more focused. The formatting is also hopefully something that the authors can make some changes to; the current formatting is not clear.

It is hoped that the authors can carefully check the data and standardise the format of the manuscript, which is very important for scientific research.

Reviewer 2 Report

The paper is interesting both for academicians and practitioners and the topic is timely. It could be considered for possible publication after the following corrections are addressed:

1. The aim of the study is presented in the abstract and methodology. The authors should repeat it in the introduction as well.

2. The scientific gap should be presented in the introduction and authors should explain how their research fulfils the highlighted gap.

3. As the journal is focused on sustainability and sustainable development, the authors should explain how the digital economy is related to sustainability.

4. The literature review section is well-developed and could be left as it is. The authors cite the articles published within the last 5 years, so it shows their acquaintance with the recent works in the field.

5. The table, which presents all the variables used in the study with their measurements and sources, from which they were extracted should be provided in the methodology.

6. all in all the methodology part is well-developed.

7. The title of the chapter 4 is "Empirical test and analysis". I would suggest changing it to "Empirical results and analysis".

8. The discussion section is missing. the authors should develop it. In that section, the authors have to provide a comparison between the research they conducted with other pieces of research conducted by other authors. 

9. Conclusions should not contain bullet points. Limitations of the study should be included in the conclusions.

Reviewer 3 Report

The presented manuscript is in the mainstream of modern scientific research within the framework of the environmental economy, and its authors aim to empirically prove the impact of the digital economy on regional carbon emissions using the example of one of the largest economies in the world. The paper is clear, relevant for the field of the green economy and presented in a well-structured manner. The advantages of the paper are, the identification of the mechanism of the influence of the digital economy on regional carbon emissions, a multidimensional robustness test. The cited references are mostly recent publications 2022-2023 years and relevant for the field of the scientific area. The study’s findings are reproducible due to the detailed methodology section. The experimental design is appropriate to test the hypothesis. The conclusions are consistent with the evidence and arguments presented in the results section.

Despite the strengths of this manuscript, a number of comments should be made that can improve it.

1.       The novelty of the paper in the part of assessing the heterogeneity in the impact of the digital economy on the western, eastern and central regions is questioned. Li et al. (2021) concluded that the impact of the digital economy on carbon emissions has certain regional heterogeneity, which is more significant in the eastern region and has a high degree of digital economy development and low resource dependence. Authors should discuss their results and show how they differ from the results by Li et al., 2021.

Reference:

Li Y, Yang X, Ran Q, et al. Energy structure, digital economy, and carbon emissions: evidence from China. Environmental Science and 1065 Pollution Research, 2021, 28: 64606-64629.

2.       Methodologically, the authors use the variable did, which represents a dummy variable whether the territory participates in the National Big Data Comprehensive Pilot Zone pilot or not. The conclusions of the study indicate that participation in the experiment automatically reduces carbon emissions, but the results of participation and the level of development of the digital economy are not described. the authors can be invited to test their conclusions using a quantitative variable of the level of development of the digital economy.

3.       The authors should provide a detailed review of empirical studies conducted on Chinese data on the impact of the digital economy on regional carbon emissions.

4.       “This indicates that the digital economy plays a significant role in reducing carbon emissions in large cities, but its impact is not significant in small and medium-sized cities, as well as megacities”. The authors should provide an explanation and in accordance with the results of other researchers: why this pattern can be seen in large cities, but not in megacities and small towns.

5.       Methodologically, the authors should explain how the level of urbanization was assessed and how it was divided into two levels “high” and “low”.

Comments on inaccuracies in the text of the paper:

1.       The formulation of hypothesis three: “Hypothesis 3: The digital economy can contribute to the reduction of carbon emissions by attracting foreign investment”. How you can check if it's “possible” or not?

2.       It is necessary to give explanations to Table 1 why the minimum values of the last three variables are negative.

Round 2

Reviewer 1 Report

Congratulations to you all. Hard work and the comments on the manuscript were handled well.

Author Response

Thank you for the comments.

Reviewer 3 Report

The authors have done significant work to improve the paper.

Author Response

Thank you for the comments.